# Does Smartphone Addiction Fall on a Continuum of Addictive Behaviors?

**DOI:** 10.3390/ijerph17020422

**Published:** 2020-01-08

**Authors:** Sheila Yu, Steve Sussman

**Affiliations:** 1Department of Preventive Medicine, University of Southern California, 2001 N Soto St., Los Angeles, CA 90032, USA; ssussma@usc.edu; 2Department of Psychology, University of Southern California, 3620 South McClintock Ave, Los Angeles, CA 90089, USA

**Keywords:** smartphone addiction, problematic smartphone use

## Abstract

Due to the high accessibility and mobility of smartphones, widespread and pervasive smartphone use has become the social norm, exposing users to various health and other risk factors. There is, however, a debate on whether addiction to smartphone use is a valid behavioral addiction that is distinct from similar conditions, such as Internet and gaming addiction. The goal of this review is to gather and integrate up-to-date research on measures of smartphone addiction (SA) and problematic smartphone use (PSU) to better understand (a) if they are distinct from other addictions that merely use the smartphone as a medium, and (b) how the disorder(s) may fall on a continuum of addictive behaviors that at some point could be considered an addiction. A systematic literature search adapted from the Preferred Reporting Items for Systematic Reviews and Meta-Analyses (PRISMA) method was conducted to find all relevant articles on SA and PSU published between 2017 and 2019. A total of 108 articles were included in the current review. Most studies neither distinguished SA from other technological addictions nor clarified whether SA was an addiction to the actual smartphone device or to the features that the device offers. Most studies also did not directly base their research on a theory to explain the etiologic origins or causal pathways of SA and its associations. Suggestions are made regarding how to address SA as an emerging behavioral addiction.

## 1. Introduction

After the iPhone entered global markets in 2007, significant technological advancements for smartphones have been made [1]. Smartphones offer a myriad of information sources at the touch of one’s fingertips and enhance productivity (e.g., synced calendars, email, alerts and alarms, global positioning system (GPS) maps), as well as provide instant access to entertainment and social networking sites and social media channels (e.g., Facebook, Twitter, Instagram) [1,2]. Due to the high accessibility and mobility of smartphones, widespread and pervasive smartphone use has become the social norm [2], which has resulted in an increase in potential distractions that expose users to various health and other risk factors (e.g., distracted driving or walking, resulting in traffic accidents; physical health detriments, such as neck and shoulder issues, blurred vision, and wrist pain; sleep disturbances; highly sedentary habits; poor academic performance; mental health concerns; financial burdens) [1,2,3,4,5].

One other consequence of excessive smartphone use is the possibility of developing an addiction. The concept of addiction historically has focused on recreational drug use, exhibiting classic characteristics related to repetitive use (i.e., tolerance, withdrawal symptoms [4,5], dependence, social problems [5], and loss of control [4]). Addiction now includes behaviors in addition to substances [1,2,3,4,5], such as gambling disorder and Internet gambling disorder [4], as discussed in the latest fifth edition of the Diagnostic and Statistical Manual of Mental Disorders (DSM–5) [4,5,6] and in the proposed International Classification of Diseases (ICD)-11 for Mortality and Morbidity Statistics [7]. The recognition of these conditions, however, did not occur immediately, as evidence of neurobiological and psychological mechanisms indicating the addictive nature of these behavioral disorders, and accurate assessment of them, took time [8].

The acknowledgement of PSU [3] has led to a debate on whether the concept of SA is a true addiction and, if it is, if it is distinct from other technological addictions engaged in on the smartphone, such as Internet and gaming addiction. Due to the recency of this phenomenon, there has been a paucity of research on its assessment [4,5]. Assessment of SA should involve reliable instruments (e.g., good internal consistency), and demonstrate adequate content validity as a measure of addiction (attempts at appetitive need fulfillment, preoccupation, loss of control, undesired consequences [9]). In addition, such a measure should capture unique consequences of SA (e.g., use while in social situations that interfere with a flow of conversation or upsets a speaker, use while driving). Limitations of knowledge in this new arena of SA/PSU studies include self-report bias from participants, lack of standardized criteria to measure SA/PSU, and different contexts that need to be considered, which may impose variation in external demands for smartphone use (e.g., sociocultural, professional, social, and academic conflicts) [10].

While there are a variety of limitations, the growing empirical research generally seems to support the concept of SA as a genuine addictive disorder, despite the available data being insufficient in establishing a valid and definitive conceptualization of SA [4]. We propose that SA may fall along a continuum of dysregulated behavior, which is problematic and even incapacitating at some point [8,11]. It is important to explore how authors to date have characterized excessive smartphone misuse and understand how various factors gleaned from current research coalesce to discern a clearer picture of this emerging behavioral addiction.

### Research Aims

The goal of this review was to examine the most recent studies of SA/PSU to see the extent to which there appears to be unique features of SA/PSU and how studies assess this behavior, to arrive at an understanding regarding at which point there is likely to be consensus on the condition being an addiction along a spectrum of problematic behaviors.

We also explored proneness to SA/PSU as a function of gender, as some researchers have suggested that SA may be more prevalent among females [12,13,14,15,16,17,18,19,20,21,22,23]. Other research has suggested that subsets of SA (e.g., social media addiction versus gaming addiction) may differentiate males from females in their smartphone use [12,14,15,24,25,26,27,28,29,30,31,32,33,34,35,36].

Additionally, we examined variation as a function of age. Some researchers have suggested that SA may be more prevalent among youth and young adults as compared to older adults and elderly who use smartphones less often than youth [12,13,19,26,30,32,34,35,37,38,39,40,41,42,43,44,45,46,47,48,49,50,51,52,53,54,55].

## 2. Materials and Methods

We conducted a systematic review of the literature on and related to SA/PSU utilizing the Preferred Reporting Items for Systematic Reviews and Meta-Analyses (PRISMA) statement [56] as a general guide. Electronic databases utilized included: MEDLINE/PubMed, ProQuest, GALE, Web of Science, Directory of Open Access Journals, Elsevier’s collection, Taylor & Francis Online, Wiley Online Library, SpringerLink, and Sage Journal’s collection.

### 2.1. Inclusion Criteria

We utilized the following search terms: “(smartphone OR “mobile phone”) AND (addiction OR “problematic use”)”, published in peer-reviewed journals between 2017 and 2019. Examining Google Scholar, we noticed a steep influx of SA/PSU articles published beginning 2017 (from 22 appearing in 2007; 41 in 2011; 164 in 2013; and 1020 in 2017). Therefore, we began the search from 2017 since this would best capture the current state of research in the area of SA/PSU research. Of the 170 articles returned from the search, 100 were retained for the current review after screening their titles and abstracts. The references of these articles were also searched, of which eight articles were deemed relevant and were included, allowing us to review 108 articles in total.

### 2.2. Exclusion Criteria

Articles published in a language other than English, duplicate articles (primary articles were retained), studies that did not substantively extend a parent study (i.e., only adapted established SA/PSU scales to a new language with some validation testing), and irrelevant articles (determined after looking through the abstracts or text; examples of irrelevant studies included those assessing the efficacy of an app serving as an intervention, focusing on the risk factors of SA/PSU rather than SA/PSU itself, and measuring invariance among SA-related scales) were excluded from the review pool (see Figure 1).

### 2.3. Information Synthesis

Since most of the literature on smartphone use were based upon SA and PSU scales, we created a table to compile scales used to measure SA/PSU of smartphones, examining the following (based on Bianchi and Phillips, 2008 [3]): (1) Scale/measurement items, (2) Cronbach’s alpha coefficients, and (3) variables and/or factors measured to assess smartphone use. A total of 14 relevant assessments emerged from a review of previous SA/PSU studies (see Table 1).

We compared the items across these assessments to see which were common among the scales (see Table 2). Based on Griffiths’ (2005) six-component model of addiction, we categorized the measured items based on the components argued to be needed “to be present for a behavior to be operationally defined as addictive”: Salience (activity becomes most important, dominating aspect), mood modification (subjective experience of feeling better, whether calming down and/or feeling less destressed), tolerance (more activity needed to feel activity’s previous effect), withdrawal (abrupt termination of activity results in unpleasant feeling, psychologically and/or physiologically), conflict (activity causes conflict between addict and others or ones’ self-image), and relapse (addictive activity patterns recur) [57].

We also extracted the following information from the studies (see Table 3): (1) First author; (2) year published; (3) whether the study classified smartphone use as an addiction (SA) or problematic smartphone use (PSU); (4) sample size (n); (5) the scale used to assess SA/PSU; (6) if SA/PSU was clearly defined; (7) if studies mentioned the negative attributes and/or risk factors of SA/PSU; (8) if studies distinguished SA/PSU from other technological addictions (e.g., Internet and gaming addiction); (9) if studies differentiated if the addiction is to the actual phone device or to what the phone device offers, such as functions, services, and/or content (e.g., social networking services, apps, etc.); and (10) if studies were structured around a theory or model to explain SA/PSU and how it affects the smartphone user. Under the scale column “adapted”, scales refer to scales adapted from SA/PSU assessment scales used in previous studies (e.g., authors compiled *x* items from scale *A* and *y* items from scale *B* to create their own scale, *S*) and/or input from field experts.

To supplement Table 3, we conducted an in-depth analysis on the study samples (see Table 4), specifically (1) the group population, either child/youth (in primary, elementary, middle, or high school), university student (students in college, graduate, and vocational/technical/institute schools were all categorized as university students), or adult; (2) sample size, n; (3) percentage of males in the sample; (4) mean age and age range of participants; (5) participant nationality; (6) sociodemographic characteristics (SDCs) (biometrics, such as body mass index (BMI), height, and weight; number of children; education; income; marital status; region or residence, such as rural or urban; and work/employment status); and (7) reported study biases. SDC data were reported for university students and adults, usually the parents of children.

The point of Table 4 was to determine the generalizability of the studies’ results: Can the research findings and conclusions of a particular group of interest be extended to the population at large? If a particular population sample is studied across age and nationality, does that imply that that group is at-risk for SA?

To supplement Table 4 on the characteristics of the studies’ samples (refer to Table 5), we examined the quality of the data collection process by noting if samples were selected as a convenience sample or randomly selected and if a sample size calculation was calculated before participant recruitment. We also categorized the data collected as consequences pertaining to: Academics (grades, academic performance), physiological (headache, eye/neck/hand pain), psychological (experiencing depression, loneliness, cravings), social (peer relationships), or usage; and whether it was objective or subjective (such as an objective clinical diagnosis of depression or a subjective self-report of experiencing depression).

Additionally, we assessed the operationalization of theory by noting if the SA/PSU scale used in the study met Griffiths’ criteria for behavioral addictions: Conflict, mood modification, relapse, salience, tolerance, and withdrawal. The 2004 Surgeon General’s report, *The Health Consequences of Smoking*, developed a framework for interpreting evidence, specifying a four-level hierarchy for interpreting evidence: (a) Evidence is sufficient to infer a causal relationship (multiple scientifically supported evidence), (b) evidence is suggestive but not sufficient to infer a causal relationship (scientifically supported evidence), (c) evidence is inadequate to infer the presence or absence of a causal relationship (evidence that is not scientifically supported and/or is sparse, of poor quality, or conflicting), or (d) evidence is suggestive of no causal relationship (no evidence) [58]. We also noted if the assumption of addiction, referencing back to the title assumption of whether the authors viewed excessive smartphone use as SA or PSU, influenced the study results.

## 3. Results

### 3.1. The Scales Used to Assess SA/PSU and Their Internal Consistency

The majority of the studies in the current review (96 out of 108) fell under the SA category (89%). Excluding one study that did not use a scale, there were nine major scales used among the studies, in order of frequency: Smartphone Addiction Scale Short Version (SAS-SV) (26%), Smartphone Addiction Proneness Scale (SAPS) (13%), Smartphone Addiction Scale (SAS) (9%), Mobile Phone Addiction Index (MPAI) (8%), Mobile Phone Addiction Scale (MPAS) (5%), Mobile Phone Problem Use Scale (MPPUS) (4%), Smartphone Addiction Scale (SPAS) (4%), Smartphone Addiction Inventory (SPAI) (3%), and SPAI Short Form (SPAI-SF) (2%). Overall, most studies either used SAS or SAS-SV (35%) or some adapted form of an SA/PSU-assessment scale based on scales from previous studies and/or input from field experts (26%). Excluding the adapted scale studies, 8 of the 12 PSU studies (67%) used SA scales to assess problematic use instead of a PSU scale, and 3 of the 95 SA studies (3%) used PSU scales to assess SA instead of SA scales (refer to Table 3).

In terms of meeting Griffiths’ criteria for behavioral addictions, all of the major scales met conflict, withdrawal, tolerance, and salience. Six of the nine met mood modification and only three met relapse (refer to Table 2). Excluding nine studies in which specific adapted scale items were unavailable, one study that did not use a scale, and 79 that used one of the nine major scales, the remaining 19 studies used an adapted SA/PSU scale that met at least half of Griffiths’ criteria for behavioral addictions (mean of 4.79, SD of 0.92): One study (5% of 19) met three criteria, seven (37%) met four, six (32%) met five, and five (26%) met all six.

All scales demonstrated validity: All scales provided a Cronbach’s alpha (CA) value as a measure of internal consistency, and performed factor structure analyses, ensuring that factor fit properly explains correlations among outcomes [59,60]. Other evaluations of validity also included testing the concurrent validity of the scale with other established scales. The CA of the nine major scales ranged from 0.83 to 0.97, with a mean of 0.90 and standard deviation (SD) of 0.05 (refer to Table 1).

### 3.2. How Studies Defined SA/PSU

In total, 87 of the 108 studies (81%) attempted a working definition of SA/PSU, of which 56 explicitly stated the definition in the paper (64%), while the rest simply characterized SA/PSU by its negative risk factors or listed associated traits. Nearly all of the studies (96%) mentioned the negative risk factors associated with SA/PSU, including physical (insomnia, pain in neck and wrists, eye soreness, etc.) and physiological concerns (depression, anxiety, loneliness, etc.). SA has been defined as a maladaptive dependency on and/or obsessive-compulsive use of the smartphone device [18], a state of being immersed in uncontrollable smartphone usage [41], and the inability to properly regulate smartphone usage to the point of experiencing adverse consequences in one’s daily life (Billieux, 2012) [21]. Interestingly, PSU has been similarly defined as “an excessive or uncontrolled use of smartphone” (also Billieux, 2012) [61]. It is interesting to note that the same reference has been used to define both SA and PSU, highlighting the interchangeability of the two terms.

Overall, 64 out of 108 studies (59%) mentioned other technological addictions, such as Internet and gaming addiction, in their paper. Of the 64 studies, only 13 (20%) explicitly argued that SA is separate and distinct from Internet addiction (2 out of 4 PSU studies; 11 out of 47 SA studies). In total, 20 studies out of 108 (19%) (4 out of 13 PSU studies; 16 out of 95 SA studies) mentioned whether SA is an addiction to the actual smartphone device or to the functions/services/content that the smartphone offers users, such as social networking services and various apps that are usually not found on other devices (refer to Table 3).

### 3.3. Theories Adopted Across These Studies

More than half of the studies (63%) did not explicitly base their study upon a specific established theory (refer to Table 3). Nine of 12 (75%) PSU studies used a theory (five of nine, 42%) or a concept, model, or hypothesis (four of nine, 33%) to guide their research; while 31 of 96 SA studies (29%) used a theory (23 of 96, 24%) or a concept or model (eight of 96, 8%) to guide their study (refer to Table 3).

The following four theories in the current review stood out as guiding points on understanding SA and how its pathways may work. First, social influence theory (Kelman,1974) helps to illustrate how socially influential factors predict a user’s intended and actual behavior in virtual settings, which is affected by three social processes: “Compliance (normative influence from others’ expectations), internalization (congruence of one’s goals with others’ goals), and identification (conception of one’s self in terms of the group’s defining features)” [39]. This theory could help to explain SA among impressionable youths, who are more prone to peer pressure to what their peers may deem cool, which could include engaging in addictive behaviors on the smartphone (e.g., games, social media, streaming videos on Twitch), as well as among adults whose lifestyles are affected by today’s heavily connected environment, which makes smartphone use a social norm.

The second theory, the theory of reasoned action (Fishbein & Ajzen, 1975), explains that a person’s actual behavior is determined by their intention to behave in a certain way, being influenced both by their own attitudes and the social context. The attitudes towards the smartphone (positive, negative, fear of missing out, and task switching) were considered key predictors of addiction [62]. This perspective focuses on the intrapersonal reasons a person may fall into SA, with positive attitudes potentially bringing about positive reinforcement to keep engaging in the cycle of excessive smartphone use. Other theories helping to illustrate user adoption and acceptance of information technologies include diffusion of innovations (Rogers, 1962), the theory of planned behavior (Ajzen, 1985), and the technology acceptance model (TAM) (Davis, 1989) [25].

However, the previous theories are limited in explaining how attitudes, perceptions, and beliefs are shaped around information technologies. To address this conceptual shortfall, cognitive absorption (CA) (Agarwal & Karahanna, 2000) was proposed as a motivating factor of usage behavior through “cognitive complexity beliefs”. CA was defined as “a state of deep involvement with software… where highly engaging and engrossing experiences result in users’ ‘deep attention’ and complete immersion and engagement with an activity” [25]. The multidimensional construct of CA has five dimensions: Temporal dissociation, focused immersion, heightened enjoyment, control, and curiosity, which follow closely with Griffiths’ biopsychosocial model of addiction.

Another relevant theory, the uses and gratifications theory (UGT), “helps understand background characteristics and individual differences motivating people to choose using *particular* types of mass media. UGT can explain how people with certain types of psychological and/or demographic characteristics may be drawn to increasingly use specific types of smartphone features” to achieve gratification [63]. Why are some people more prone to using Instagram and obsessing over how many “likes” they have accrued on their posts? What is it about some people having to constantly check their Twitter to see if their “tweets” are trending or how many “retweets” they have received? This theory could play a role in defining and characterizing social media addiction as a subset of SA.

### 3.4. Group-Specific Variation in SA

Of the total 108 studies, 84 (78%) focused on adolescents or emerging adults (ranging from elementary, middle, and high school students (32 of 84 studies, 38%) to college/university students (50 of 84, 59%) (the remaining two studies looked at both high school and university students)) and mentioned that smartphone use was highly prevalent among this youth/young adult group as compared to a more elderly group. Fourteen studies (13% of 108) focused on adults, and 10 studies included both child and adult participants, usually family members (10% of 108) (refer to Table 4).

Nineteen studies (18%) reported that gender was a significant predictor of SA, specifically that being female was significantly associated with higher tendencies toward SA or smartphone dependency or that SA/dependency/risk for SA was more prevalent in females than in males. Sixteen studies (15%) suggested that females may be more likely to be addicted to social media while males may be more likely to be addicted to gaming. However, no consistent findings regarding the subtypes of SA by gender were uncovered.

### 3.5. Generalizability of Study Results and Biases

Across all studies (refer to Table 4), the average percent of male participants was 46% (SD = 13). Twenty-four countries were represented in this review, with 90 studies (83%) categorized as a South, Southeast, or East Asian country (in order of frequency: South Korea, China, Turkey, Taiwan, India, Saudi Arabia, Lebanon, Bangladesh, Hong Kong, Indonesia, Iran, Iraq, and Malaysia).

All studies reported gender and age, and seven reported ethnicity (six studies with American adult participants and one with Brazilian adult participants). We also considered other common sociodemographic characteristics (SDCs) measures, including biometrics (e.g., height, weight, and BMI); number of children; education, income, marital, and work/employment status; and region (northern, southern, western, or eastern) or residence (urban or rural). Fifty of the 108 studies (46%) did not measure any SDCs, 27 (25%) looked at 1 SDC, 18 (17%) looked at 2, 9 (8%) looked at 3, and 4 (4%) looked at 4 SDCs.

About a third of the studies (35%) reported sampling bias (from convenience sampling), which did not allow for generalization of the findings outside of the study group population. For example, the results from studies focusing on university students may not apply to the general public. Recall bias of participants’ self-reports on actual smartphone use and/or dependence, information bias, social desirability bias, response bias, and selection bias were also noted. Thirty-nine of the studies (36%) also reported multiple biases while 17 (16%) did not report any biases.

Only two studies used longitudinal data [64,65], and none of the others conducted follow-up assessments on the cross-sectional analyses of smartphone use and behavior. That is, limitations included: The use of cross-sectional data, which limits the ability to draw causal inferences, especially when determining the direction of association between SA/PSU and risk factors of interest; small sample size; not being able to determine whether study characteristics preceded SA/PSU development or were the outcome of smartphone use; use of less-than-optimal instruments tending to be subjective rather than objective; incentive-influenced survey answers; and attrition. The studies that did not report biases or limitations may have overlooked noting the potential ones mentioned above.

Nearly all of the authors recommended one or more of the following: Conducting future studies to further investigate the relationship between SA/PSU and related health risks, thoroughly identifying positive and negative outcomes, conducting longitudinally designed research studies with broader sample profiles; creating public health educational programs to inform the public of the physical and psychological risks associated with SA/PSU, and developing proper evidence-based strategies and interventions to address SA/PSU.

### 3.6. Quality of Samples and Level of Evidence

Examining data collection methods (refer to Table 5), about a half of the studies (49%) reported using convenience sampling, about a third (34%) reported random sampling, and the rest (17%) did not clearly report a sampling method. Most of the studies (90%) did not calculate an appropriate sample size before recruiting participants. In terms of sample sizes (with a range of 52–7003), 17 studies (16%) had 200 participants or less, 41 (38%) fell in the 201–500 range, 24 (22%) in the 501–1000 range, and 26 (24%) had more than 1001 participants.

In terms of associations with SA/PSU, 1 study focused on academic data, 11 (10%) presented physiological data (e.g., craniovertebral angle, skin conductance), 40 (37%) psychological data (e.g., depression, anxiety), 16 (15%) social data (e.g., social connectedness, alexithymia), 8 (7%) smartphone usage data (e.g., duration, types of functions used), and 28 (26%) reported a mix of those categories.

Most of the studies were subjective reports (90%). Ten of the 11 objective reports (one academic, four physiological, one psychological, two social, and two smartphone usage) were deemed suggestive (but not sufficient or indicative) of a causal association with SA/PSU. The remaining objective report (psychological) was deemed insufficient because the link between SA and national identity was not clear to us. Seven of the 97 subjective reports (six social, one psychological) were deemed suggestive (but not sufficient or indicative) because unlike other psychological reports (within which subjective reports can be verified with objective testing, or physiological measures), certain social aspects (e.g., loneliness, family history of alcohol addiction, need for social assurance, life satisfaction, parental neglect, friends’ support, and peer relationships) are hard to measure with objective testing and so subjective data is helpful and could be relatively valid and reliable in those cases.

## 4. Discussion

### 4.1. Smartphone Addiction on a Continuum of Addictive Behaviors

#### 4.1.1. Is Smartphone Addiction Really an Entity of Its Own?

With smartphone use engrained as the social norm today, and the pervasive use ever increasing despite an awareness of the health risks and adverse consequences, now the question is if excessive smartphone use is truly an addiction (SA) or just problematic smartphone use (PSU). This review highlights that SA articles already assume that SA is an addiction and frame the research as such, while PSU articles explore reasons why PSU falls short of meeting the necessary criteria to be considered a true addiction. The growing literature on excessive smartphone use has conceptualized the disorder as an addictive behavior. Specifically, advocates of PSU argue that the ethiopathological pathways and processes have not yet been identified in SA research, suggesting that SA interventions are simply targeting the symptoms rather than the underlying causes [66]. We agree that research on SA etiology is necessary in order for the disorder to be properly and accurately diagnosed. Although that will take a considerable amount of time and effort to accrue, hopefully our future capabilities to capture sound psychosocial as well as neurobiological evidence will be established. There has also been criticism that the only support to date of SA being an addiction is limited to “exploratory studies relying on self-report data which is collected via convenience sampl[ing]” [4]. Although the majority of recent reports have been subjective, there have been attempts to collect objective data that are promising, especially if those studies are planning on following up on the data in the future. It seems more studies are also trying to carry out randomized studies with larger sample sizes in order to expand on the empirical data available in the field of SA research.

The current review found that all of the assessment scales measuring SA/PSU met at least half of Griffiths’ criteria, which need to be met in order for a disorder to be considered an addiction, according to Griffiths. However, PSU advocates counter that conforming excessive smartphone use within addiction models, such as Griffiths’, could oversimplify the disorder and result in clinical irrelevance [66]. Specifically, attempts at conceptualizing tolerance with respect to SA may be insufficient, as inferring tolerance based on the increasing use of the smartphone could vary by several factors, such as age (e.g., teens pressured by peers to participate in social media use), subscription status (subscriptions to apps are paid in full or need to be paid monthly), relationship status (single versus in a relationship), occupation status (student, works desk job), and significant life events (starting or ending a romantic relationship) [4]. Although it could indeed oversimply the condition, Griffiths’ model has been widely used as a biopsychosocial framework to operationalize addictive components, and so it is a good starting point to conceptualize the level of addiction to the smartphone. Also, once again, in order to truly verify conceptualizations like tolerance, there is a need for neurobiological evidence (e.g., “alteration/sensitization in specific cerebral circuitries” [4]) to confirm tolerance levels increasing in a smartphone user.

We considered using the “Surgeon General’s criteria” (the “Hill criteria”) for causality, noting the: (1) Consistency, (2) strength, (3) specificity, (4) temporal relationship, and (5) coherence of the association between SA/PSU and the variables of interest. However, we chose not to include consistency, as none of the studies conducted follow-up studies in different populations under different circumstances, or specificity, which researchers have criticized to be useless or misleading [67]. However, the vast majority of the studies only met the coherence criteria. Additionally, since all but two studies were longitudinal, temporality could not be assessed. Future studies could consider addressing the consistency, strength, and temporal relationship of the association by conducting longitudinal and follow-up studies, with adequate sample sizes ensuring power in the analyses.

#### 4.1.2. Assuming SA Is an Addiction, Is It an Addiction to the Device or on the Device?

What used to be a novelty-type activity about a decade ago has now become more of a normative behavior [68]; excessive smartphone use is one of the more recent forms of human–machine interaction, raising public health alarms. However, the current review notes that despite the growing body of research, many studies still do not clearly define SA. In fact, in some of the literature, SA and PSU are still interchangeable terms. To add to the ambiguity, only about one in five articles make the distinction that SA is an addiction specifically to the mobile features provided by the smartphone (texting; various social media apps, such as Twitter and Instagram) that desktops and even laptops cannot match in terms of ease of portability and handheld capabilities. The vast majority of recent articles did not make a clear distinction of SA from related addictions, such as Internet or gaming addiction, suggesting that researchers may assume that SA is actually a subset of technological addictions. Those articles focused more on the mediating role of associated variables (e.g., emotional intelligence and coping style [48]) on SA and/or the moderating role of variables on the relation between risk factors of SA and SA (e.g., whether perceived social support and depression would moderate the relationship between sensation seeking and adolescent SA [49]) rather than differentiating SA from other technological addictions or specifying that SA is an addiction distinct from other addictions (e.g., gaming, gambling, shopping, socializing, sex) based on what the smartphone offers users.

We believe that the only difference between smartphone and Internet addiction (IA) is that SA is essentially IA presented through a highly portable device. IA is a potential addiction restricted to a stationary type of device. In contrast, excessive smartphone use is more prevalent due to its ease of access and portability, and the smartphone itself has become the most common device of choice for people today to access the Internet. Currently, the smartphone is the medium of problematic overuse of apps, games, and social networking site interactions. In the future, perhaps a new behavioral addiction will be virtual reality addiction through use of high-tech contact lens—the ever-changing and improving technological advancements will make possibilities (and potential problems) endless. However, for today, we argue that SA is an emerging addiction to the smartphone content specifically through the smartphone device.

#### 4.1.3. Generalizability of Results and At-Risk Populations

The emergence of SA as an addiction is highlighted by the fact that people from various countries are affected by SA, which is not localized to just one continent but worldwide. Although several factors (e.g., not all sociodemographic characteristics of samples being measured, biases, cross-sectional limitations of inferencing causality, insufficient sample sizes, inability to determine directionality of SA and its risk factors, attrition) did not allow for generalization of the findings outside of the study group population, the results still suggested that children and young adults are more affected by SA than other age groups and that men and women are affected by SA in different ways.

In terms of SA affecting youth, one study reported that those at risk of developing SA displayed more severe levels of behavioral and emotional problems, lower self-esteem, and poorer quality of communication with their parents compared with those at normal risk [45]. In terms of gender differences, one study observed that girls who frequently use their smartphones may have a greater tendency to use social networking apps (e.g., Facebook) to upload pictures/share their lives online and therefore have a higher degree of smartphone attachment as compared to boys. Girls have also been reported to form and maintain social relationships and be engaged emotionally through constant app connection while boys mostly use smartphones to communicate through texts [36]. However, not all people at risk fall neatly into those observed categories. Implications of these differences require further research among similar group populations (e.g., youths, females versus males, students) around the world to better understand age, gender, and cultural variations in SA that may exist, which could help guide future tailored interventions.

### 4.2. Limitations

For this review, we narrowed our article pool by searching for only “smartphone”/“mobile phone” and “addiction”/“problematic use” terms. We did not actively search for “android”, “iPhone”, “cell phone”, “cellular device”, “compulsive use”, “phubbing”, “snubbing”, and “nomophobia”, which could have potentially made us miss articles relevant to this review. However, we were able to locate a few relevant articles after reviewing the reference sections of selected articles. Much more research is needed that investigates differences as a function of demographic variables, such as gender, age, or ethnicity. Another limitation is that our interpretations of categorizing certain variables into their respective groupings may not match others’ opinions. For example, while we may propose that a certain subjective report is suggestive of a causal relationship with SA/PSU, another researcher may disagree and say that it is insufficient evidence. Another example would be missing a guiding theory used in articles, making Table 3 incomplete.

## 5. Conclusions

Most studies to date seem to assume that SA is a valid behavioral addiction, many forming their assumptions based on Griffiths’ component model of addiction, and framing their study based on that assumption. The interchangeable use of the terms SA and PSU, inconsistent methodological approaches used to study SA (e.g., varied use of SA/PSU scales among research), lack of standardized diagnostic criteria, and unclear distinctions of SA from other related addictions make it difficult to make a conclusive statement on the status of SA, which could be considered “an ill-defined and heterogeneous construct” [67]. With no unifying theory on SA, all theories mentioned in the current research highlight the complexity of SA: One theory cannot simply explain SA, but rather several theories and models possibly need to be integrated to better explain its distinct addictive traits in this new technologically advanced era. Much more research is needed to confirm the uniqueness of SA, which encompasses the addictive activities engaged in on the smartphone, which includes apps that are not available on other devices. It is most plausible, based on the current studies, to infer that SA falls on a continuum of additive behaviors, from mild PSU to more extreme addictive behavior, where the consequences need to be addressed, prevented, and potentially treated before the adverse health effects debilitate the smartphone user.

## Figures and Tables

**Figure 1 ijerph-17-00422-f001:**
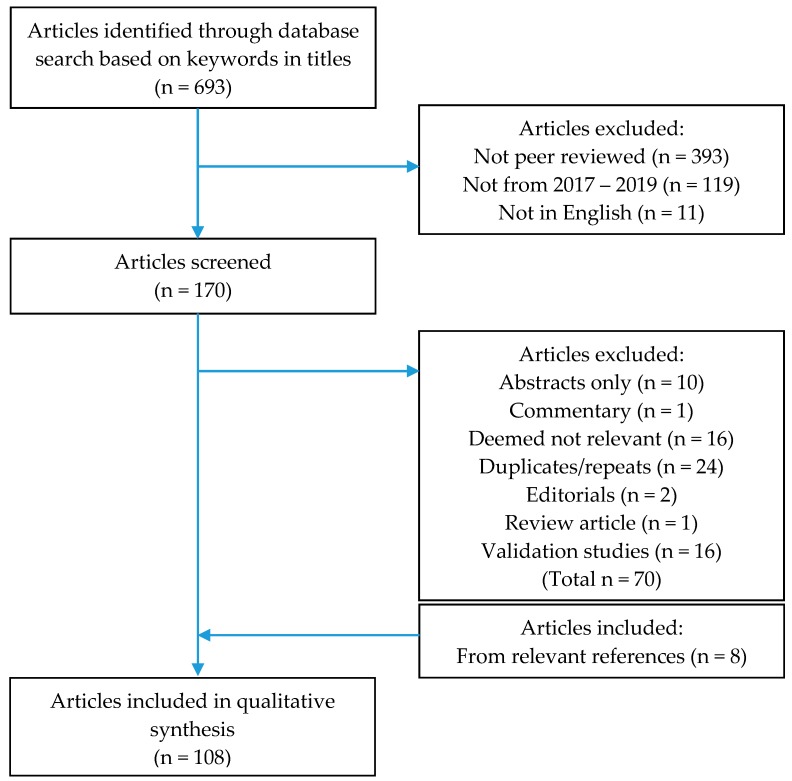
Process of final article selection for the current review.

**Table 1 ijerph-17-00422-t001:** Smartphone addiction/problematic smartphone use assessments in the literature.

Title (Abbreviation): Authors	Year Cat ^1^	Items CA ^2^	Measures/Factors
Mobile Phone Problem Use Scale (MPPUS): Bianchi and Phillips	2005 PSU	27 0.93	1. Tolerance
			2. Escape from other problems
			3. Withdrawal
			4. Craving
			5. Negative life consequences
Mobile Phone Addiction Index (MPAI): Leung	2008 SA	17 0.87	1. Inability to control craving
			2. Feeling anxious and lost
			3. Withdrawal or escape
			4. Productivity loss
Problematic Mobile Phone Use Questionnaire (PMPUQ): Billieux, Van der Linden, and Rochat	2008 PSU	30 0.77	1. Dangerous/prohibited use
			2. Dependence
			3. Financial problems
Smart Mobile Phone Addiction Scale (MPAS): Hong, Chiu, and Huang	2012 SA	11 0.90	1. Academic problems and effects
			2. Time management and problems
			3. Substitute satisfaction
Problematic Use of Mobile Phones (PUMP) Scale: Merlo, Stone, and Bibbey	2013 PSU	20 0.94	1. Tolerance
			2. Withdrawal
			3. Longer time than intended
			4. Great deal of time spent
			5. Craving
			6. Activities given up or reduced
			7. Use despite physical/psychological problems
			8. Failure to fulfill role obligations
			9. Use in physically hazardous situations
			10. Use despite social/interpersonal problems
Smartphone Addiction Scale (SAS): Kwon, Lee, Won, Park, Min, Hahn, Gu, Choi, and Kim	2013 SA	48 0.97	1. Daily-life disturbance
			2. Positive anticipation
			3. Withdrawal
			4. Cyberspace-oriented relationship
			5. Overuse
			6. Tolerance
Short Version for Adolescents (SAS-SV): Kwon, Kim, Cho, Yang	2013 SA	10 0.91	1. Daily-life disturbance
			2. Withdrawal
			3. Cyberspace-oriented relationship
			4. Overuse
			5. Tolerance
Smartphone Addiction Scale for College Students (SAS-C): Su, Pan, Liu, Chen, Wang, and Li	2014 SA	22 0.93	1. Withdrawal behavior
			2. Salience behavior
			3. Social comfort
			4. Negative effects
			5. Use of application (app)
			6. Renewal of app
Smartphone Addiction Management System (SAMS): Lee, Ahn, Choi, and Choi	2014 SA	14 N/A	1. Daily-life disturbance
			2. Virtual world orientation
			3. Withdrawal
			4. Tolerance
Smartphone Addiction Inventory (SPAI): Lin, Chang, Lee, Tseng, Kuo, and Chen	2014 SA	26 0.94	1. Compulsive behavior
			2. Functional impairment
			3. Withdrawal
			4. Tolerance
Korean Smartphone Addiction Proneness Scale (SAPS ^3^): Kim, Lee, Lee, Nam, and Chung	2014 SA	29 0.88	1. Disturbance of adaptive functions
			2. Withdrawal
			3. Tolerance
			4. Virtual life orientation
Smartphone Addiction Scale (SPAS): Bian and Leung	2014 SA	19 0.83	1. Disregard of harmful consequences
			2. Preoccupation
			3. Inability to control craving
			4. Productivity loss
			5. Feeling anxious and lost
Mobile Phone Problem Use Scale: derivation of a short scale (MPPUS-10): Foerster, Roser, Schoeni, and Röösli	2015 PSU	10 0.85	1. Craving
			2. Withdrawal
			3. Dependence
			4. Loss of control
			5. Negative life consequences
Short form of Smartphone Addiction Inventory (SPAI-SF): Lin, Pan, Lin, and Chen	2016 SA	10 0.84	1. Compulsive behavior
			2. Functional impairment
			3. Withdrawal
			4. Tolerance

Notes. ^1^ Category: Smartphone addiction (SA) or problematic smartphone use (PSU); ^2^ Cronbach’s alpha value; ^3^ First developed by Korea’s National Information Society Agency (2011) “Report on the Development of Korean Smartphone Addiction Proneness Scale for Youth and Adults” (also known as S-Scale) [Korean].

**Table 2 ijerph-17-00422-t002:** Comparison of major validated scale variables based on Griffiths’ component model.

	MPPUS	MPAI	MPAS	SAS	SAS-SV	SPAI	SPAI-SF	SAPS	SPAS	Total
Conflict	√	√	√	√	√	√	√	√	√	9
Withdrawal	√	√	√	√	√	√	√	√	√	9
Tolerance	√	√	√	√	√	√	√	√	√	9
Salience	√	√	√	√	√	√	√	√	√	9
Mood	√	√		√		√		√	√	6
Relapse						√	√	√		3

**Table 3 ijerph-17-00422-t003:** Characteristics of relevant smartphone studies, 2017–2019.

First Author	Year	Cat ^1^	Scale	Define ^2^	Distinct ^3^	D v. F ^4^	Theory
Abed	2018	SA	Adapted	No	No	No	n/a
Akodu	2018	SA	SAS-SV	No	No	Yes	n/a
Akturk	2018	SA	SAS-SV	Yes	No	No	n/a
AlAbdulwahab	2017	SA	SAS	No	No	No	n/a
Alavi	2018	SA	MPPUS	No	No	No	Yes ^5^
Albursan	2019	SA	Adapted	No	Yes	Yes	n/a
Alhassan	2018	SA	SAS-SV	Yes	No	No	n/a
Alhazmi	2018	SA	SAS-SV	Yes	No	No	n/a
Arefin	2017	SA	Adapted	Yes	Yes	Yes	n/a
Arnavut	2018	SA	SAS-SV	Yes	No	No	n/a
Barnes	2019	SA	Adapted	Yes	Yes	Yes	Yes ^6^
Basu	2018	SA	Adapted	Yes	No	No	n/a
Beison	2017	PSU	MPPUS	No	No	No	Yes ^7^
Cerit	2018	SA	SAS	Yes	No	No	n/a
Cha	2018	SA	SAPS	Yes	No	No	n/a
Chang	2019	SA	SPAI	Yes	No	No	n/a
Chen	2018	SA	Adapted	No	Yes	Yes	Yes ^8^
Chen	2017	SA	Adapted	Yes	No	No	Yes ^9^
Chen	2017	SA	SAS-SV	Yes	No	No	n/a
Chiang	2019	SA	SPAI-SF	No	No	No	n/a
Cho	2017	SA	SAPS	Yes	No	No	Yes ^10^
Cho	2017	SA	AMT ^11^	No	No	No	n/a
Choi	2017	SA	SAS	Yes	No	No	n/a
Chou	2019	SA	SPAI-SF	No	No	No	Yes ^12^
Chung	2018	SA	SAPS	No	No	No	n/a
Cocoradă	2018	SA	SAS-SV	Yes	No	No	Yes ^13^
Contractor	2017	PSU	SAS-SV	Yes	No	No	Yes ^14^
De-Sola	2017	SA	MPPUS	Yes	Yes	No	n/a
Ding	2019	PSU	SOCS	No	No	Yes	n/a
Duke	2017	SA	SAS-SV	No	Yes	Yes	n/a
Elhai	2017	PSU	SAS	No	No	No	Yes ^15^
Elserty	2018	SA	SAS-SV	Yes	No	No	n/a
Gao	2018	SA	MPAI	Yes	No	No	Yes ^16^
Gao	2017	SA	MPAS	No	No	No	n/a
Gezgin	2018	SA	SAS-SV	Yes	No	No	n/a
Gligor	2019	SA	MPDQ ^17^	No	No	No	n/a
Gökçearslan	2018	SA	SAS-SV	Yes	No	No	n/a
Habibi	2018	SA	Adapted	Yes	No	No	n/a
Han	2017	SA	MPAI	Yes	Yes	No	Yes ^18^
Hao	2019	SA	MPAI	Yes	No	No	n/a
Hawi	2017	SA	SAS-SV	No	No	No	n/a
Heo	2018	SA	SAPS	No	No	No	n/a
Herrero	2019	SA	SPAS	No	No	No	Yes ^10^
Herrero	2019	SA	SPAS	Yes	No	No	n/a
Herrero	2019	SA	SPAS	Yes	No	No	n/a
Ihm	2018	SA	Adapted	No	No	No	n/a
Jeong	2019	SA	SAPS	Yes	Yes	No	n/a
Jin	2017	SA	SAS-SV	Yes	No	No	Yes ^18,19^
Jo	2018	SA	SAPS	No	No	No	n/a
Khoury	2019	SA	SPAI	No	Yes	No	n/a
Kim	2018	PSU	Adapted	Yes	No	No	Yes ^20^
Kim	2017	PSU	Adapted	Yes	No	No	n/a
Kim	2019	SA	SAPS	No	No	No	n/a
Kim	2018	SA	SAPS	Yes	No	No	n/a
Kim	2017	SA	SAPS	No	No	No	Yes ^18^
Kim	2017	SA	SAPS	Yes	No	No	Yes ^14^
Kim	2017	SA	SAS	No	No	No	n/a
Kim	2018	SA	SAPS	Yes	No	No	Yes ^18^
Kita	2018	SA	SAS-SV	Yes	No	No	n/a
Konan	2019	SA	SAS-SV	Yes	No	No	n/a
Konan	2018	SA	SAS-SV	No	No	No	n/a
Kuang-Tsan	2017	SA	MPAS	No	No	No	n/a
Kumcağız	2019	SA	SAS-SV	No	No	No	n/a
Kumcağız	2017	SA	SAS-SV	No	No	No	n/a
Kuss	2018	PSU	PMPUQ	Yes	No	No	Yes ^21^
Kwak	2018	SA	Adapted	No	No	No	Yes ^22^
Kwan	2017	SA	SAS	Yes	No	Yes	Yes ^23,24^
Lan	2018	SA	MPIAS ^25^	No	No	No	n/a
Lee	2018	PSU	SAPS	Yes	No	No	Yes ^26^
Lee	2017	SA	SAPS	Yes	Yes	No	n/a
Lee	2017	SA	SAPS	No	No	No	n/a
Lee	2017	SA	Adapted	No	No	No	n/a
Lee	2018	SA	Adapted	No	No	No	n/a
Lee	2018	SA	Adapted	No	No	No	n/a
Lee	2018	SA	SAS-SV	No	Yes	No	n/a
Lee	2018	SA	SAS-SV	Yes	Yes	No	Yes ^27^
Li	2018	SA	SPAS	No	No	No	Yes ^28^
Lian	2018	SA	MPAI	Yes	No	No	Yes ^26,29^
Lian	2017	SA	MPAI	Yes	No	No	Yes ^26^
Lin	2017	SA	App	Yes	Yes	Yes	n/a
Liu	2018	SA	MPAI	Yes	No	No	Yes ^22^
Liu	2018	PSU	SAS-SV	Yes	No	Yes	Yes ^30^
Liu	2017	SA	MPAI	No	No	No	Yes ^31^
Lu	2018	SA	MPAS	No	No	No	n/a
Mahapatra	2018	SA	Adapted	Yes	No	Yes	Yes ^32^
Matar B.	2017	SA	SPAI	No	Yes	No	n/a
Megna	2018	SA	SAS-SV	Yes	No	No	n/a
Mei	2018	SA	MPAI	Yes	No	Yes	Yes ^33^
Nayak	2018	SA	Adapted	Yes	Yes	No	n/a
Noë	2019	SA	SAS	Yes	Yes	Yes	n/a
Parasuraman	2017	SA	Adapted	No	No	Yes	n/a
Salvi	2018	SA	Adapted	Yes	No	No	n/a
Sekhon	2018	SA	MPPUS	Yes	No	No	n/a
Serin	2019	SA	SAS-SV	Yes	No	No	n/a
Sok	2019	SA	Adapted	No	No	No	n/a
Song	2018	SA	SAS (Lee)	No	No	No	n/a
Sun	2019	SA	SAS (Ku)	Yes	No	No	Yes ^34^
Tunc-Aksan	2019	SA	SAS	No	No	Yes	Yes ^35^
Wang	2018	SA	SAS-SV	No	No	No	Yes ^36,37^
Wang	2017	SA	SAS-SV	No	No	No	Yes ^38^
Wolniewicz	2018	PSU	SAS-SV	Yes	Yes	Yes	Yes ^15,16^
Xu	2019	PSU	MPAS	Yes	No	No	Yes ^22,39^
Yang	2019	SA	MPAI	Yes	No	No	Yes ^40^
Yang	2019	PSU	Adapted	Yes	No	No	n/a
Yildiz Durak	2017	SA	SAS	Yes	Yes	No	Yes ^41^
Yildiz	2017	SA	SAS-SV	No	No	No	n/a
You	2019	SA	MPAS	No	No	No	Yes ^42^
Youn	2018	SA	SAS	No	No	No	n/a

Notes. ^1^ Category based on study title: Smartphone addiction (SA) or problematic smartphone use (PSU); ^2^ Define: If SA or PSU was defined; ^3^ Distinct: If studies distinguished SA from other technological addictions (e.g., Internet, gaming addiction); ^4^ D v. F: If studies differentiated if the addiction is to the actual phone device or to what the phone device offers, such as functions, services, and/or content (e.g., social networking services, apps, etc.); ^5^ Psychosocial Theory of Development (Erikson, 1963) and Adolescent Identity Paradigm (Marcia, 1991); ^6^ Cognitive absorption (Agarwal & Karahanna, 2000); ^7^ Reward Deficiency Syndrome hypothesis (Blum et al., 1996); ^8^ Social influence theory (Rashotte, 2007); ^9^ Four categories of drinking motives (Stewart & Devine, 2000); ^10^ Big Five personality traits (Norman, 1963); ^11^ Addiction Measurement Tools of Measuring Smartphone Addiction of Children-Adolescents (Korea Network Information Center); ^12^ 6-T Internet attitude model (Chou, Wu, & Chen, 2013); ^13^ Theory of Reasoned Action (Fishbein & Ajzen, 1975); ^14^ Impulsive pathway perspective (Billieux, 2012); ^15^ Uses and Gratifications Theory (UGT) (Blumler & Katz, 1974); ^16^ Compensatory Internet Use Theory (CIUT) (Kardefelt-Winther, 2014); ^17^ MPDQ: Mobile phone dependence questionnaire; ^18^ Attachment theory (Bowlby, 1969); ^19^ Psychoanalytic theory (Kassel et al., 2007); ^20^ Social enhancement model (Kraut et al., 2002); ^21^ Biopsychosocial model of addiction (Griffiths, 2005); ^22^ General strain theory (Agnew, 1992); ^23^ Theory of parenting style (Baumrind, 1971); ^24^ First theory of self-regulation (Asgari et al., 2011); ^25^ MPIAS: Mobile Phone Internet Addiction Scale; ^26^ Problem behavior theory (PBT) (Jessor, 1977); ^27^ Power distance belief (Hofstede, 1980); ^28^ Media system dependency (MSD) theory (Ball-Rokeach & DeFleur, 1976); ^29^ Social skills deficit theory (Valkenburg & Peter, 2007); ^30^ Social compensation theory (Zell & Moeller, 2018); ^31^ Response style theory (Nolen-Hoeksema, 1991); ^32^ Incentive-Sensitization (Robinson & Berridge, 2003) & Learning Theory (Wallace, 1999); ^33^ Hierarchy of needs theory (Maslow, 1968); ^34^ Risky families model (Repetti, 2002); ^35^ Intrinsic motivation theory (Przybylski, Weinstein, Ryan, & Rigby, 2009); ^36^ Sensation seeking theory (Zuckerman, 1994); ^37^ Social support buffering hypothesis (Cohen & Wills, 1985); ^38^ Cognitive-behavioral model (Davis, 2001); ^39^ Resilience theory (Fergus & Zimmerman, 2005); ^40^ Diathesis-stress theories (Monroe & Simons, 1991) and stress-buffering hypothesis (Cohen & Edwards, 1989); ^41^ Social Cognitive Theory (Bandura, 1986); ^42^ Sociometer theory of self-esteem (Leary et al., 1995).

**Table 4 ijerph-17-00422-t004:** Demographic profile of smartphone study samples and reported study biases.

First Author	Pop ^1^	n ^2^	Males (%)	Age (m (r) ^3^)	Nationality	SDC ^4^ (BCEIMRW)	Reported Biases
Abed	Univ	229	35	n/a	Iraqi	(BCEIMRW)	None
Akodu	Univ	77	57	22	Nigerian	(BCEIMRW)	Sampling
Akturk	Univ	1156	49	n/a	Turkish	(BCEIMRW)	Sampling
AlAbdulwahab	Univ	78	50	21	Saudi	(BCEIMRW)	None
Alavi	Univ	500	21	28 (18–31+)	Iranian	(BCEIMRW)	Sampling
Albursan	Univ	2008	45	22 (17–28)	Arab		Sampling
Alhassan	Adult	935	34	32 (18–55+)	Saudi	(BCEIMRW)	Multiple
Alhazmi	Univ	181	48	24	Saudi	(BCEIMRW)	Sampling
Arefin	Univ	247	54	(18–27)	Bangladeshi		Sampling
Arnavut	Adult	714	58	(18–30+)	Turkish		Sampling
Barnes	Univ	140	31	(18–35+)	American	(BCEIMRW)	Multiple
Basu	Univ	388	60	20	Indian	(BCEIMRW)	Sampling
Beison	Univ	100	25	20 (18–23)	American ^5^	(BCEIMRW)	None
Cerit	Univ	214	20	20 (18–26)	Turkish		Recall
Cha	MS	1824	51	16	Korean	(BCEIMRW)	Multiple
Chang	ES/Adult	5089	52/31	n/a/43	Taiwanese	(BCEIMRW)	Multiple
Chen	Univ	2000	49	21 (17–23)	Taiwanese		Multiple
Chen	Univ	384	54	n/a	Chinese		Multiple
Chen	Univ	1441	48	20 (17–26)	Chinese	(BCEIMRW)	Multiple
Chiang	ES/MS	2155	52	n/a	Taiwanese	(BCEIMRW)	Multiple
Cho	Adult	400	52	(20–40+)	Korean	(BCEIMRW)	Multiple
Cho	PS/Adult	303	7/51	(20–40+)/(0–6)	Korean	(BCEIMRW)	Sampling
Choi	HS	1020	52	n/a	Korean	(BCEIMRW)	Sampling
Chou	HS/Adult	1444	43/n/a	n/a/n/a	Taiwanese	(BCEIMRW)	Recall
Chung	MS/HS	1796	46	15 (13–18)	Korean		Sampling
Cocoradă	HS/Univ	717	35	20	Romanian		Multiple
Contractor	Adult	346	42	34	American **^5^**	(BCEIMRW)	Multiple
De-Sola	Adult	1126	48	33 (16–65)	Spanish	(BCEIMRW)	Multiple
Ding	Univ	849	56	n/a	Chinese		Multiple
Duke	Adult	262	36	32	German		Recall
Elhai	Adult	308	54	33	American **^5^**	(BCEIMRW)	Multiple
Elserty	Univ	420	32	20	Egyptian	(BCEIMRW)	None
Gao	Univ	1105	29	21 (16–25)	Chinese	(BCEIMRW)	Multiple
Gao	Univ	722	48	20 (15–24)	Chinese	(BCEIMRW)	Multiple
Gezgin	HS	161	58	16	Turkish		Multiple
Gligor	Univ	150	44	27	Romanian	(BCEIMRW)	Multiple
Gökçearslan	Univ	885	41	n/a	Turkish	(BCEIMRW)	None
Habibi	HS	271	n/a	17	Indonesian		None
Han	Univ	543	41	20 (17–22)	Chinese		Sampling
Hao	Univ	847	51	20 (18–24)	Chinese	(BCEIMRW)	Sampling
Hawi	Univ	381	59	21 (17–27)	Lebanese		Multiple
Heo	HS	790	23	n/a	Korean		None
Herrero	All	526	52	(15–55+)	Spanish	(BCEIMRW)	None
Herrero	All	241	55	(15–55+)	Spanish	(BCEIMRW)	Sampling
Herrero	All	416	52	(15–55+)	Spanish	(BCEIMRW)	Selection
Ihm	Youth	2000	50	12	Korean		Recall
Jeong	HS	768	58	n/a	Korean	(BCEIMRW)	Multiple
Jin	Univ	297	55	20 (17–24)	Chinese	(BCEIMRW)	Sampling
Jo	All	7003	45	(14–39)	Korean	(BCEIMRW)	Recall
Khoury	Univ	100	48	(18–25)	Brazilian ^5^	(BCEIMRW)	Response
Kim	Adult	615	51	30 (19–40)	American ^5^		Recall
Kim	All	930	52	26 (13–40)	American ^5^	(BCEIMRW)	Recall
Kim	MS/HS	4512	45	15	Korean	(BCEIMRW)	Multiple
Kim	Youth	3380	51	(10–19)	Korean	(BCEIMRW)	Information
Kim	Univ	200	37	22 (19–28)	Korean		Sampling
Kim	Univ	608	30	23	Korean	(BCEIMRW)	Multiple
Kim	HS	1479	48	n/a	Korean	(BCEIMRW)	Sampling
Kim	Univ	313	42	22 (17–29)	Korean		Sampling
Kita	YA	221	65	19 (17–22)	Israeli		Sampling
Konan	Univ	496	25	n/a	Turkish		None
Konan	Univ	330	36	(20–24)	Turkish		None
Kuang-Tsan	Univ	332	65	(18–22)	Taiwanese		Sampling
Kumcağız	HS	352	44	16 (14–19)	Turkish	(BCEIMRW)	Multiple
Kumcağız	Adult	428	37	40 (21–65)	Turkish	(BCEIMRW)	Sampling
Kuss	All	273	26	28 (16–65)	Various		None
Kwak	MS	1170	42	n/a	Korean	(BCEIMRW)	Sampling
Kwan	Univ	211	35	22	Hong Kong	(BCEIMRW)	None
Lan	Univ	1044	48	21	Chinese		Information
Lee	Youth	231	40	16 (13–18)	Korean	(BCEIMRW)	Sampling
Lee	MS	370	49	13	Korean		Multiple
Lee	MS/HS	3000	53	n/a	Korean		Sampling
Lee	MS/HS	1125	51	n/a	Korean	(BCEIMRW)	Multiple
Lee	Univ	125	49	n/a	Korean		None
Lee	Univ	324	9	n/a	Korean		Multiple
Lee	MS	490	100	14	Korean	(BCEIMRW)	Multiple
Lee	Adult	778	63/58 ^6^	35/25 ^6^	Various	(BCEIMRW)	Information
Li	Adult	527	46	27 (18–35)	Chinese	(BCEIMRW)	Sampling
Lian	Univ	716	54	20 (18–24)	Chinese	(BCEIMRW)	Sampling
Lian	Univ	682	58	19 (18–24)	Chinese		Recall
Lin	Univ	79	72	22	Taiwanese		Sampling
Liu	HS	899	46	17 (14–19)	Chinese		Sampling
Liu	Univ	465	31	19 (16–24)	Chinese		Sampling
Liu	HS	1196	53	17 (14–20)	Chinese		Multiple
Lu	MS	1311	54	15	Various		Sampling
Mahapatra	HS/Univ	330	58	(15–20)	Indian		Multiple
Matar B.	Univ	688	53	21	Lebanese	(BCEIMRW)	Recall
Megna	Adult	52	46	27 (18–35)	Italian		None
Mei	Univ	1034	47	20	Chinese	(BCEIMRW)	Multiple
Nayak	Univ	429	35	20 (16–29)	Indian		None
Noë	Adult	64	53	25 (19–46)	British	(BCEIMRW)	Sampling
Parasuraman	Adult	409	42	23 (18–55)	Malaysian	(BCEIMRW)	Sampling
Salvi	Univ	100	59	21 (18–25)	Indian		None
Sekhon	Univ	80	50	(20–24)	Indian		Sampling
Serin	Univ	287	14	n/a	Turkish	(BCEIMRW)	Multiple
Sok	Univ	139	16	n/a	Korean		Sampling
Song	PS/Adult	328	n/a/0	(3–5)/n/a	Korean	(BCEIMRW)	Multiple
Sun	HS	1041	56	12 (11–15)	Chinese		Multiple
Tunc-Aksan	HS	296	54	n/a	Turkish	(BCEIMRW)	None
Wang	MS	655	55	17 (15–19)	Chinese		Multiple
Wang	MS	768	44	17 (15–19)	Chinese	(BCEIMRW)	Multiple
Wolniewicz	Univ	296	43	20	American ^5^	(BCEIMRW)	Multiple
Xu	MS	316	47	14 (12–16)	Chinese		Sampling
Yang	HS	1258	53	17 (14–20)	Chinese		Sampling
Yang	Univ	218	58	18 (16–19+)	Taiwanese		Multiple
Yildiz Durak	MS/HS	612	52	13 (10–18)	Turkish	(BCEIMRW)	Multiple
Yildiz	HS	262	50	17 (14–19)	Turkish		Sampling
You	Univ	653	50	20 (17–25)	Chinese	(BCEIMRW)	Sampling
Youn	Youth	158	53	15 (12–19)	Korean	(BCEIMRW)	Multiple

Notes. ^1^ Pop: Population general group (PS: Primary school; ES: Elementary school; MS: Middle school; HS: High school; Univ: University, college, or institute of technology students; YA: Young adult); ^2^ Sample size n; ^3^ m (r): mean age (range, if available); ^4^ SDC: Common sociodemographic characteristics besides gender, age, and ethnicity (B = biometrics: height, weight, BMI; C = number of children; E = education; I = income; M = marital status; R = regional/residence: urban or rural; W: work/employment status); ^5^ Reports racial status (e.g., African-American/Black, American Indian, Asian, Native Hawaiian/other Pacific Islander, White) and ethnicity (e.g., Hispanic/Latino, Not Hispanic/Latino); ^6^ Of 778 total, 431 US and 347 Chinese participants: male % and mean age of US and Chinese, respectively.

**Table 5 ijerph-17-00422-t005:** The quality of samples, level of evidence, and theoretical framework of the studies.

First Author	Sel ^1^	SSS ^2^	Evidence ^3^	Obj ^4^	Int ^5^	Griffiths ^6^	Inf ^7^
Abed	R	No	PhPs	S	I	n/a	Yes
Akodu	C	No	Ph	O	Sg	CMRSTW	Yes
Akturk	C	Yes	S	S	Sg	CMRSTW	Yes
AlAbdulwahab	C	Yes	Ph	S	I	CMRSTW	Yes
Alavi	C	Yes	Ps	O	I	CMRSTW	Yes
Albursan	R	No	Ps	S	I	n/a	Yes
Alhassan	R	No	Ps	S	I	CMRSTW	Yes
Alhazmi	R	Yes	Ph	S	I	CMRSTW	Yes
Arefin	C	No	A	O	Sg	CMRSTW	Yes
Arnavut	R	No	U	S	I	CMRSTW	Yes
Barnes	C	No	Ps	S	I	CMRSTW	Yes
Basu	R	Yes	U	S	I	CMRSTW	Yes
Beison	n/a	No	S	S	Sg	CMRSTW	Yes
Cerit	R	Yes	Ph	S	I	CMRSTW	Yes
Cha	R	No	PhPsU	S	I	CMRSTW	Yes
Chang	R	No	SU	S	I	CMRSTW	Yes
Chen	C	No	PsS	S	I	n/a	PSU ^8^
Chen	C	No	Ps	S	I	CMRSTW	Yes
Chen	R	No	PhPs	S	I	CMRSTW	Yes
Chiang	R	No	Ps	S	I	CMRSTW	Yes
Cho	R	No	Ps	S	I	CMRSTW	Yes
Cho	R	No	Ps	S	I	n/a	Yes
Choi	n/a	No	U	S	I	CMRSTW	Yes
Chou	R	No	U	S	I	CMRSTW	Yes
Chung	R	No	Ph	S	I	CMRSTW	Yes
Cocoradă	C	No	PsU	S	I	CMRSTW	Yes
Contractor	C	No	Ps	S	I	CMRSTW	Yes
De-Sola	C	No	Ps	S	I	CMRSTW	Yes
Ding	R	No	PsU	S	I	CMRSTW	Yes
Duke	C	No	U	S	I	CMRSTW	Yes
Elhai	C	No	PsU	S	I	CMRSTW	Yes
Elserty	C	No	PhU	S	I	CMRSTW	Yes
Gao	C	No	Ps	S	I	CMRSTW	Yes
Gao	C	No	Ps	S	I	CMRSTW	Yes
Gezgin	C	No	PsU	S	I	CMRSTW	Yes
Gligor	n/a	No	Ps	S	I	CMRSTW	Yes
Gökçearslan	C	No	PsS	S	I	CMRSTW	Yes
Habibi	n/a	No	Ps	S	I	CMRSTW	Yes
Han	R	No	Ps	S	I	CMRSTW	Yes
Hao	R	No	PsU	S	I	CMRSTW	Yes
Hawi	R	No	Ps	S	I	CMRSTW	Yes
Heo	C	Yes	Ps	S	I	CMRSTW	Yes
Herrero	R	No	PsSU	S	I	CMRSTW	Yes
Herrero	R	No	Ps	S	I	CMRSTW	Yes
Herrero	R	No	Ps	S	I	CMRSTW	Yes
Ihm	R	No	S	S	I	CMRSTW	Yes
Jeong	n/a	No	PsS	S	I	CMRSTW	Yes
Jin	C	No	Ps	PsS	I	CMRSTW	Yes
Jo	n/a	No	Ps	S	I	CMRSTW	Yes
Khoury	R	No	Ph	O	Sg	CMRSTW	Yes
Kim	R	No	Ps	S	I	CMRSTW	Yes
Kim	R	No	Ps	S	I	CMRSTW	Yes
Kim	R	No	Ps	S	I	CMRSTW	Yes
Kim	R	No	S	S	I	CMRSTW	Yes
Kim	n/a	Yes	Ps	S	I	CMRSTW	Yes
Kim	C	No	Ph	S	Sg	CMRSTW	Yes
Kim	n/a	No	Ph	S	I	CMRSTW	Yes
Kim	C	No	Ps	S	I	CMRSTW	Yes
Kita	C	No	S	O	Sg	CMRSTW	Yes
Konan	R	No	PsS	S	I	CMRSTW	Yes
Konan	C	No	Ps	S	I	CMRSTW	Yes
Kuang-Tsan	C	No	S	S	Sg	CMRSTW	Yes
Kumcağız	C	No	S	S	I	CMRSTW	Yes
Kumcağız	C	No	S	S	I	CMRSTW	Yes
Kuss	C	No	Ps	S	I	CMRSTW	Yes
Kwak	C	No	PsS	S	Sg	CMRSTW	Yes
Kwan	C	No	PsS	S	I	CMRSTW	Yes
Lan	R	No	Ps	S	Sg	n/a	Yes
Lee	C	Yes	PsS	S	Sg	CMRSTW	Yes
Lee	C	No	Ps	O	Sg	CMRSTW	PSU ^8^
Lee	n/a	No	S	S	Sg	CMRSTW	Yes
Lee	R	No	Ph	S	I	CMRSTW	Yes
Lee	R	No	S	O	Sg	CMRSTW	Yes
Lee	C	No	S	S	I	CMRSTW	Yes
Lee	n/a	No	Ps	S	I	CMRSTW	Yes
Lee	n/a	No	Ps	S	I	CMRSTW	Yes
Li	C	No	S	S	I	CMRSTW	Yes
Lian	C	No	S	S	I	CMRSTW	Yes
Lian	C	No	S	S	I	CMRSTW	Yes
Lin	n/a	No	U	O	Sg	n/a	Yes
Liu	R	No	Ps	S	I	CMRSTW	Yes
Liu	C	No	PsS	S	I	CMRSTW	Yes
Liu	C	No	PhPs	S	I	CMRSTW	Yes
Lu	C	No	S	S	I	CMRSTW	Yes
Mahapatra	C	No	APsS	S	I	CMRSTW	Yes
Matar B.	R	No	PsU	S	I	CMRSTW	Yes
Megna	C	No	Ph	O	Sg	CMRSTW	Yes
Mei	C	Yes	Ps	S	I	CMRSTW	Yes
Nayak	R	No	AU	S	I	CMRSTW	Yes
Noë	C	No	U	O	Sg	CMRSTW	Yes
Parasuraman	C	No	U	S	I	n/a	Yes
Salvi	n/a	No	Ph	O	Sg	n/a	Yes
Sekhon	n/a	No	Ps	S	I	CMRSTW	Yes
Serin	C	No	Ps	S	I	CMRSTW	Yes
Sok	C	Yes	PsSU	S	I	n/a	Yes
Song	C	No	S	S	I	CMRSTW	Yes
Sun	n/a	No	PsS	S	I	n/a	Yes
Tunc-Aksan	R	No	APsS	S	I	CMRSTW	Yes
Wang	n/a	No	PsS	S	I	CMRSTW	Yes
Wang	n/a	No	PsS	S	I	CMRSTW	Yes
Wolniewicz	C	No	PsU	S	I	CMRSTW	Yes
Xu	n/a	No	APs	S	I	CMRSTW	Yes
Yang	R	No	Ps	S	I	CMRSTW	Yes
Yang	C	No	PhPsU	S	I	CMRSTW	Yes
Yildiz Durak	C	No	Ps	S	I	CMRSTW	Yes
Yildiz	C	No	Ps	S	I	CSTW	Yes
You	C	No	Ps	S	I	CMRSTW	Yes
Youn	C	No	PsS	S	I	CMRSTW	Yes

Notes. ^1^ Sel: Selection of sample: C = convenience sample, R = randomly selected; ^2^ SSS: Sufficiency of sample size: Yes or No; ^3^ Evidence: Type of data presented: A = academic, Ph = physiological, Ps = psychological, S = social, U = usage; ^4^ Obj: Objectivity: O = objective, S = subjective; ^5^ IE: Interpretation of the level of evidence: Sf = sufficient, Sg = suggestive but not sufficient, I = inadequate, NR = suggestive of no causal relationship; ^6^ Griffiths: C = conflict, M = mood modification, R = relapse, S = salience, T = tolerance, W = withdrawal; ^7^ Inf: Assumption of addiction influencing study results: Yes or No; ^8^ Was titled an SA study but conclusions use PSU terminology.

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
