# Peer review of "Does Smartphone Addiction Fall on a Continuum of Addictive Behaviors?"

_ijerph, 2020, doi:10.3390/ijerph17020422_

Round 1

Reviewer 1 Report

It is a well-constructed and developed systematic review, following the PRISMA criteria.

However, the results section is too reduced, leaving the reader a little disappointed. There is a need for more in-depth analysis of the selected articles, stating the level of evidence and recommendation of each of them, the quality of the samples developed, the representativeness of the samples, the potential biases and other aspects related to the degree of generalisation of the results.

It is possible that, as the authors mention, the very disparity of the studies makes it difficult to obtain valid conclusions to continue the investigation of these problems, but this was a previously known problem that precisely motivated the revision and cannot be an excuse to limit the analysis of results. Specifically, the title announces that what is going to be studied is the possible existence of a behavioural continuum between smartphone abuse and addiction, but no analysis reasons whether such a continuum exists or not, limiting itself to providing percentages of those who take one or another view of the problem. To such an extent, the research question has not been answered.

It would be desirable to know to what extent the previous positioning in certain theories or ateoricity, the previous assumption of addiction or mere abuse and the rest of the characteristics studied influence or condition the results, to what extent the instruments used respond to the previous theoretical positions and other questions that may inform the objective of the research. Otherwise, all the effort made to carry out the systematic review translates into a poverty of findings that is not capable of giving an answer to the question posed.

We believe that the review is relevant and well designed, but the results obtained respond to a poor analysis that fails to find answers to the research question posed from the title.

Author Response

We truly thank you for your invaluable feedback and believe that your expert comments have improved the manuscript. Please see the attachment.

Reviewer 2 Report

This paper is greatly needed in this field and I commend the authors for their efforts on consolidating this vastly developing literature to devise some important conclusions about SA. Overall, the authors make clear statements about their methods, exclusion/inclusion decisions and report the findings in a useful way based on the behavioural addiction framework. This is very helpful to synthesise this literature to establish the correspondence across studies in this regard. I do not have any recommendations which I feel need to be made for this to be deemed publishable. Thank you to the authors for their efforts here and this worthy contribution to the literature. 

Author Response

Thank you so much for your support, we truly appreciate your time and efforts!

Reviewer 3 Report

This paper is a meta-analysis on relation of smartphone addiction and other behavioral addictions. 

A SLR was conducted to gather information from existing researches to examine whether the SA/PSU fall on continuum of common addictions. 

The main findings suggested that it was not conclusive yet as main definitions had not been unified and many literature were biases. 

While there are some detailed reports on what was found in literature, which might be useful to some extent, the work seemed to be preliminary.

English needs polish as well as there are many typos errors.

Author Response

Thank you for your feedback, please see the attachment below.

Round 2

Reviewer 1 Report

The authors have responded to the suggestions and implemented the changes, which are acceptable. In the opinion of this reviewer, the article can be accepted for publication.

Reviewer 3 Report

Authors tried best to address our earlier comments. I think the revised version Claire’s many issues and took a down to earth approach to reflect the current status quo in the field concerned. Now it becomes a worthwhile contribution to the journal. On the other hand, English needs further polish